# NSCLC EGFR Mutation Prediction via Random Forest Model: A Clinical–CT–Radiomics Integration Approach

**DOI:** 10.3390/arm93050039

**Published:** 2025-09-26

**Authors:** Anass Benfares, Badreddine Alami, Sara Boukansa, Mamoun Qjidaa, Ikram Benomar, Mounia Serraj, Ahmed Lakhssassi, Mohammed Ouazzani Jamil, Mustapha Maaroufi, Hassan Qjidaa

**Affiliations:** 1Faculty of Sciences, Department of Computer Science, Sidi Mohammed Ben Abdellah University, Fez 30000, Morocco; sara.boukansa@usmba.ac.ma (S.B.); ikram.benomar@usmba.ac.ma (I.B.); 2Faculty of Medicine, Department of Radiology, Sidi Mohammed Ben Abdellah University, Fez 30000, Morocco; badreddine.alami@usmba.ac.ma (B.A.); mounia.serraj@usmba.ac.ma (M.S.); dr_mstph@yahoo.fr (M.M.); 3Faculty of Sciences, Department of Computer Science, Mohammed V University, Rabat 10090, Morocco; qmamoun@gmail.com; 4Department of Computer Science and Engineering, Université du Québec en Outaouais, Gatineau, QC J8X 3X7, Canada; ahmed.lakhssassi@uqo.ca; 5Faculty of Engineering Sciences, Private University of Fez, Fez 30000, Morocco; ouazzani@upf.ac.ma

**Keywords:** EGFR mutation, non-small cell lung cancer (NSCLC), radiomics, CT imaging, machine learning, Random Forest, SHAP, feature selection, precision oncology, non-invasive diagnosis

## Abstract

**Highlights:**

**What are the main findings?**
Accurate estimation of epidermal growth factor receptor (EGFR) mutation status in NSCLC patients can be achieved through a predictive framework combining clinical, CT, and radiomic information.The best-performing Random Forest model (11 features) achieved an AUC of 0.91 (95% CI: 0.81–1.00). Subgroup results were EGFR-WT (F1-score = 0.91 ± 0.02) and EGFR-Mutant (F1-score = 0.68 ± 0.04), confirming balanced though differentiated predictive performance.

**What is the implication of the main finding?**
The proposed non-invasive prediction tool may assist in early identification of candidates for tyrosine kinase inhibitor (TKI) therapy when tissue sampling is limited.This integrative approach supports the development of AI-driven, personalized diagnostic strategies in lung cancer management.

**Abstract:**

Non-small cell lung cancer (NSCLC) is the leading cause of cancer-related mortality worldwide. Accurate determination of epidermal growth factor receptor (EGFR) mutation status is essential for selecting patients eligible for tyrosine kinase inhibitors (TKIs). However, invasive genotyping is often limited by tissue accessibility and sample quality. This study presents a non-invasive machine learning model combining clinical data, CT morphological features, and radiomic descriptors to predict EGFR mutation status. A retrospective cohort of 138 patients with confirmed EGFR status and pre-treatment CT scans was analyzed. Radiomic features were extracted with PyRadiomics, and feature selection applied mutual information, Spearman correlation, and wrapper-based methods. Five Random Forest models were trained with different feature sets. The best-performing model, based on 11 selected variables, achieved an AUC of 0.91 (95% CI: 0.81–1.00) under stratified five-fold cross-validation, with an accuracy of 0.88 ± 0.03. Subgroup analysis showed that EGFR-WT had a performance of precision 0.93 ± 0.04, recall 0.92 ± 0.03, F1-score 0.91 ± 0.02, and EGFR-Mutant had a performance of precision 0.76 ± 0.05, recall 0.71 ± 0.05, F1-score 0.68 ± 0.04. SHapley Additive exPlanations (SHAP) analysis identified tobacco use, enhancement pattern, and gray-level-zone entropy as key predictors. Decision curve analysis confirmed clinical utility, supporting its role as a non-invasive tool for EGFR-screening.

## 1. Introduction

Lung cancer continues to represent the deadliest form of cancer worldwide, accounting for nearly 2.2 million newly diagnosed cases and 1.8 million deaths in 2020. It remains the primary cause of cancer-related mortality across both sexes [1,2]. Non-small cell lung cancer (NSCLC) constitutes approximately 85% of all lung cancer diagnoses, with adenocarcinoma being the most frequently encountered histological subtype [3,4].

Within the molecular landscape of NSCLC, a considerable number of tumors harbor oncogenic driver mutations. Among these, alterations in the epidermal growth factor receptor (EGFR) gene play a pivotal role in treatment stratification [5,6,7]. These mutations are identified in 10–15% of NSCLC cases in Western populations and in up to half of patients in East Asia [8]. Two common activating mutations—exon 19 deletions and the L858R point mutation in exon 21—are predictive of tumor sensitivity to tyrosine kinase inhibitors (TKIs), including gefitinib, erlotinib, afatinib, and osimertinib [9,10,11]. The clinical benefits of these agents over traditional chemotherapy have been confirmed through several randomized clinical trials, thereby justifying their integration into global treatment guidelines [12,13,14,15,16].

Despite the availability of targeted treatments, detection of EGFR mutations typically requires molecular testing on tissue samples, often obtained via invasive biopsy. These procedures are not always feasible in patients with comorbidities or advanced disease and can be compromised by tumor heterogeneity, resulting in sampling errors [17,18,19]. Moreover, molecular profiles may evolve throughout therapy, sometimes requiring additional tissue sampling, which poses logistical and clinical challenges [20]. Although liquid biopsy using circulating tumor DNA (ctDNA) has emerged as a minimally invasive alternative, its diagnostic performance may be limited, particularly in early-stage disease or in cases with low tumor burden [21,22].

Radiomics has emerged as a promising tool for extracting high-dimensional, quantitative data from medical images, potentially revealing hidden imaging biomarkers associated with underlying genomic profiles such as EGFR mutation status [23,24,25,26,27,28]. When combined with clinical information and CT-derived morphological indicators, radiomics offers a valuable pathway toward non-invasive tumor genotyping strategies [29].

Numerous prior investigations have explored the application of machine learning algorithms for predicting EGFR mutation status, often combining radiomic features with clinical data to enhance predictive accuracy [30,31,32,33]. While these approaches have shown encouraging results, many remain limited by challenges such as lack of transparency, insufficient external validation, or the use of unenhanced CT images, which may compromise generalizability [34,35,36,37]. With the increasing demand for transparent AI in medicine, combining powerful classifiers (e.g., Random Forests, SVMs, and neural networks) with SHAP-based interpretation has shown promise in delivering explainable and effective diagnostic predictions. However, previous studies leveraging explainable machine learning in oncology have faced several limitations. Many were restricted to either handcrafted radiomic features or purely deep-learning-based pipelines, often requiring large training cohorts and providing limited interpretability in small or heterogeneous datasets. In particular, works such as [38,39,40] focused primarily on demonstrating proof-of-concept explainability without explicitly integrating multimodal clinical, morphological, and radiomic variables into a unified model. Our study addresses this gap by combining these complementary feature domains within an interpretable Random Forest framework, specifically tailored to a modest cohort size. This integration not only enhances predictive robustness but also ensures clinical interpretability through SHAP-based analysis, thereby improving the translational potential of the approach. SHAP assigns a measurable impact to each input feature, helping to demystify how complex models arrive at their decisions, thus improving clinician confidence and regulatory compliance [23].

Studies that combine radiomic features with conventional clinical and imaging variables—such as patient age, smoking history, sex, enhancement characteristics, and pleural retraction—have demonstrated superior prediction performance compared to models based on a single modality [37,41]. In situations where biopsy is unfeasible or molecular diagnostics are not readily accessible, such integrated models could help to identify patients likely to benefit from early TKI intervention and support equitable deployment of personalized treatment strategies in oncology [42,43].

Our work, in this regard, presents an explainable and high-performing machine learning strategy, designed to harness the predictive strength of selected clinical data, CT scan features, and radiomic markers for determining EGFR mutation in NSCLC patients. In this study, we specifically focused on patients with lung adenocarcinoma, as this NSCLC subtype is most strongly associated with EGFR mutations and represents the standard population for the clinical implementation of targeted therapies. Feature selection was performed using a three-step process involving mutual information, Spearman rank correlation, and the FeatureWiz algorithm, allowing for effective dimensionality reduction while preserving key predictive variables. Random Forest classifiers were employed due to their ability to capture non-linear interactions and to prevent overfitting. Model interpretability was further enhanced using SHAP analysis to visualize and understand the contribution of each feature to the model’s output.

To our knowledge, this study is one of the earliest to implement a validated radiomics-driven machine learning approach for EGFR mutation prediction in NSCLC within a North African cohort. The findings support the practical integration of artificial intelligence into oncology workflows, even in environments with constrained resources, and align with international efforts to broaden equitable access to precision medicine on a global scale.

## 2. Materials and Methods

### 2.1. Study Population and Methodological Framework

Data were retrospectively collected from 521 patients with histologically confirmed NSCLC treated at the Hassan II University Hospital. Eligibility was contingent upon the availability of a pre-therapeutic chest CT conducted within 3 months prior to biopsy or surgical intervention, a confirmed EGFR mutation status, and complete clinical data. Only patients with histologically confirmed lung adenocarcinoma were included, as this histological subtype accounts for the majority of EGFR mutations in NSCLC and ensures the biological relevance of the predictive modeling. Patients were excluded if CT data were incomplete, if imaging was compromised by artifacts, or if histology was inconsistent with adenocarcinoma. Based on these criteria, 138 patients were included: Among the entire cohort, 98 patients were identified with wild-type EGFR, while 40 patients harbored activating EGFR mutations. Model evaluation was conducted exclusively via stratified five-fold cross-validation on the full dataset (*n* = 138: EGFR-WT = 98, EGFR-Mutant = 40). All preprocessing steps and hyperparameter tuning were performed within the training portion of each fold only, with the validation portion kept unseen to prevent information leakage. The overall patient selection and allocation process is depicted in Figure 1.

A detailed summary of the dataset characteristics, including source, histology, patient distribution, and acquisition period, is presented in Table 1.

The methodological workflow followed in this study consisted of several integrated steps: acquisition of standardized high-resolution chest CT scans (120 kVp, 100–200 mAs, slice thickness 1–1.25 mm, lung and mediastinal kernels, venous contrast phase at ~70 s); tumor segmentation using semi-automatic tools with expert validation in 3D Slicer (version 5.6.2; an open-source platform for medical image segmentation and radiomics analysis) and ITK-SNAP (version 4.0.1); extraction of radiomic features with PyRadiomics (version 3.1.0), including shape, intensity, and texture metrics such as the Gray-Level Co-occurrence Matrix (GLCM), Gray-Level Run Length Matrix (GLRLM), Gray-Level Size Zone Matrix (GLSZM), Gray-Level Dependence Matrix (GLDM), and Neighborhood Gray-Tone Difference Matrix (NGTDM); feature selection via mutual information, Spearman correlation filtering (where features with r > 0.85 were considered redundant and one of them was removed to reduce collinearity), and the FeatureWiz algorithm. It is important to note that this threshold of Spearman correlation (r > 0.85) was applied only for redundancy filtering among radiomic variables, while the correlation coefficients reported later in the Results section reflect the individual association of selected features with EGFR mutation status, which are naturally more moderate, and training/evaluation of five Random Forest classifiers using different feature subsets. Performance was assessed with AUC, accuracy, precision, recall, and F1-score. Model interpretability was ensured using SHAP. An overview of the workflow is provided in Figure 2.

### 2.2. EGFR Mutation Analysis

Formalin-fixed, paraffin-embedded (FFPE) tumor tissue specimens were obtained from all patients and deemed suitable for downstream molecular analysis. Histopathological assessment was initially conducted to estimate the percentage of tumor cells within each sample. Subsequently, DNA was extracted from regions enriched in tumor cells using the QIAamp DNA FFPE Tissue Kit (Qiagen, Hilden, Germany).

Depending on tumor cellularity, one of two detection strategies was adopted. Low-cellularity samples were examined via PCR, while those with ≥30% tumor content underwent NGS analysis using the BigDye Terminator v3.1 sequencing system (Applied Biosystems, Foster City, CA, USA). EGFR mutations were subsequently classified into three categories: wild-type tumors, which showed no detectable mutations; frequently observed alterations, which included in-frame deletions in exon 19 and L858R substitutions within exon 21; and uncommon or complex mutations, such as G719X in exon 18, S768I, T790M, insertions in exons 19 or 20, and compound variants.

### 2.3. CT Imaging Protocol and Preprocessing Workflow

A multidetector CT system from Siemens (SOMATOM Definition, Erlangen, Germany) was used to acquire all thoracic CT images, adhering strictly to institutional standard imaging procedures. Raw data were stored in DICOM format and processed using open-source Python libraries, notably pydicom for metadata handling and scipy.ndimage for initial volume manipulations [44,45].

To ensure spatial uniformity, all CT images were resampled to a resolution of 226 × 226 pixels using the Elastix registration module embedded in the 3D Slicer platform (version 5.6.2). This resampling process mitigated variability stemming from acquisition parameters and enabled consistency in radiomic analysis.

Segmentation of the primary tumors was achieved through a hybrid methodology combining manual delineation and semi-automatic approaches. The segmentation was carried out using ITK-SNAP and 3D Slicer, with manual adjustments applied in regions where semi-automatic tools failed to define clear tumor boundaries, especially in peripheral or low-contrast lesions. To enhance reliability and reduce interobserver bias, all segmentations were independently reviewed and validated by two experienced thoracic radiologists, and only those reaching full agreement were retained for analysis.

This rigorous image preprocessing ensured the standardization and reproducibility of the dataset prior to radiomic feature extraction.

### 2.4. Tumor ROI Delineation and Extraction

Tumor contouring was achieved using three distinct segmentation strategies: a manual delineation method, a semi-automated segmentation with ITK-SNAP, and a supplementary semi-automatic protocol integrated into 3D Slicer [23,39,40]. Each tumor was segmented independently using these three methods. Only ROIs showing consistent boundaries across all approaches were retained as final segmentations. This multi-method validation strategy was crucial for minimizing variability and ensuring the robustness of subsequent radiomic analysis (Figure 3).

For each patient, the region of interest was defined as the entire primary lung tumor volume, excluding adjacent atelectasis, vascular structures, or pleural tissue. This ensured that the radiomic features extracted were specific to tumor tissue and avoided contamination by surrounding anatomical structures.

### 2.5. Description of Clinical, Morphological, and Radiomic Characteristics

#### 2.5.1. Demographic and Clinical Characteristics

Patient electronic medical records were retrospectively reviewed to extract clinical and demographic variables such as age, sex, smoking history, and EGFR mutation status. Prior to analysis, all variables were thoroughly reviewed to ensure data completeness, accuracy, and internal consistency across the dataset.

#### 2.5.2. CT Morphological Features

Radiomic feature extraction was performed using PyRadiomics, an open-source and extensively validated library for high-throughput quantitative medical image analysis. The process was implemented through a custom script within the SlicerRadiomics™ interface (v2.10), ensuring consistent and reproducible workflows across the dataset (accessed on 25 May 2023).

The extracted features included shape-based descriptors quantifying tumor geometry and spatial properties, first-order statistics summarizing voxel intensity distributions, and texture features derived from GLCM, GLRLM, GLSZM, and GLDM. These metrics capture complex patterns of intra-tumoral heterogeneity, reflecting spatial voxel arrangements, edge sharpness, and structural irregularities. All features were extracted from validated ROIs using standardized settings and organized into structured matrices for subsequent machine learning analysis.

#### 2.5.3. Extraction of Radiomic Features

To evaluate whether limiting the training dataset to only the most informative features could improve the predictive performance of the model, we designed five distinct datasets. These datasets were generated by systematically combining clinical, CT, and radiomic variables (either in full or after applying relevance-based feature selection).

Subsequently, a total of five Random Forest (RF) classifiers [46,47,48] were developed. Each model was trained using a unique feature combination derived from one of the five constructed datasets.

Random Forests were chosen as the core classification method owing to their robustness in managing mixed-type and high-dimensional datasets, as well as their ability to mitigate overfitting in relatively small and imbalanced cohorts. In addition, they provide transparent feature importance measures, which align with the study’s objective of developing an interpretable and clinically applicable predictive model.

All five models were trained independently to predict EGFR mutation status. The specific feature configurations used for each classifier are summarized in Table 2.

To further ensure robustness and reproducibility, all Random Forest models underwent hyperparameter tuning via a five-fold stratified cross-validated grid search. The search space included n_estimators (100 to 1000), max_depth (3 to 20), min_samples_split (2 to 10), and min_samples_leaf (1 to 5). The AUC metric was used as the optimization criterion. The final selected hyperparameters for each of the five models are summarized in Appendix A.

This comparative modeling approach allowed us to assess the relative contribution of each feature type (clinical, morphological (CT), and radiomic), as well as the impact of dimensionality reduction, through selection of the most relevant predictors.

### 2.6. Cross-Validation Framework for Model Training and Evaluation

To maintain balanced class representation and ensure reproducibility in model evaluation, we adopted a stratified five-fold cross-validation strategy. The full cohort (*n* = 138: EGFR-WT = 98; EGFR-Mutant = 40) was partitioned into five folds, each preserving the relative distribution of mutation status. In each iteration, four folds (≈80% of the data) were used for model training and one fold (≈20%) was reserved for validation. This process was repeated until every fold had served once as the validation set.

All preprocessing steps (feature selection, correlation filtering, and FeatureWiz selection) and hyperparameter tuning were performed within the training portion of each fold only, ensuring that the corresponding validation fold remained unseen, thus preventing information leakage. Performance metrics are reported as mean ± standard deviation across folds, providing more stable and reproducible estimates under class imbalance. Robustness was further verified through repeated cross-validation with different random seeds and bootstrap confidence intervals. The aggregated confusion matrices are provided in Appendix A.

#### 2.6.1. Addressing Class Imbalance and Data Limitations

Because the training dataset exhibited a marked imbalance, with a lower proportion of EGFR-mutated cases, the Synthetic Minority Over-Sampling Technique (SMOTE) was applied during preprocessing. This algorithm generates synthetic minority samples through interpolation, thereby balancing the dataset and improving the generalizability of the models. SMOTE has been widely adopted in biomedical machine learning for mitigating class imbalance, and its use here ensured a more reliable learning process.

#### 2.6.2. Data Augmentation Strategies

To further strengthen model robustness and reduce overfitting, classical image augmentation techniques were employed. These included random horizontal flipping, rotations of up to 25°, horizontal and vertical translations to simulate spatial shifts, random occlusion to introduce noise, and Gaussian perturbations to mimic intensity variability.

These augmentation strategies were deliberately chosen over more advanced approaches such as CutMix, MixUp, or CutOut. While such methods have demonstrated effectiveness in natural image classification, they may generate anatomically implausible CT slices and distort radiomic texture patterns, thereby reducing model interpretability. The selected classical transformations preserve anatomical realism and maintain the integrity of radiomic descriptors, ensuring both robustness and the clinical relevance of the predictive models.

#### 2.6.3. Summary of the Training Approach

The final training strategy therefore combined three main elements: balanced data through SMOTE, expansion of the dataset using slice-based modeling, and generalization enhancement via data augmentation. This integrated approach was specifically designed to maximize classifier performance despite the relatively limited number of available patients. In addition, the computational complexity of all Random Forest models was assessed to demonstrate the feasibility of clinical translation.

To further ensure transparency, we also assessed the computational complexity of all Random Forest models. Training and evaluation were performed on a workstation equipped with an Intel Core i7-12700 CPU, 32 GB RAM, and an NVIDIA RTX 3060 GPU. Training time ranged from 20–45 s depending on the model, with inference requiring less than 0.1 s per patient. Memory usage remained below 3 GB across all models. These results confirm the efficiency and scalability of our approach compared to more resource-intensive deep learning methods, thereby reinforcing its suitability for clinical translation.

## 3. Results

### 3.1. Cohort Composition and Mutation Status Overview

The study ultimately included 138 patients with histologically confirmed non-small cell lung cancer (NSCLC), each presenting with comprehensive clinical, imaging, and molecular data suitable for analysis. Among them, 40 patients (28.98%) were identified as harboring EGFR mutations, while the remaining 98 patients (71.01%) exhibited a wild-type EGFR profile. This distribution provided the foundation for all subsequent comparative, statistical, and predictive modeling analyses.

### 3.2. Feature Selection and Statistical Dependency Analysis

#### 3.2.1. CT, Clinical, and Radiomic Feature Relevance Selection

A structured two-stage selection procedure was implemented to determine the CT and clinical features most strongly associated with EGFR mutation status. First, mutual information analysis was performed to quantify the statistical dependency between each feature and the EGFR mutation label. This method, well-suited for capturing both linear and non-linear relationships, enabled the identification of variables with the highest discriminative potential. The mutual information scores for the top-ranked features are presented in Table 3.

In the second step, a Spearman rank correlation analysis was conducted to assess monotonic associations among the selected variables. Features showing high inter-correlation (Spearman’s *p* > 0.85) were iteratively removed, using a greedy recursive elimination strategy, in order to retain only the most independent predictors. Table 4 displays the 95% confidence intervals and corresponding *p*-values for each of the retained features.

Based on the mutual information scores presented in Table 3, the enhancement pattern (homogeneous vs. heterogeneous) was identified as the most informative variable for predicting EGFR mutation status, with a score of 0.4517. It was followed by tobacco use (0.3289) and age (0.2584), both of which demonstrated substantial predictive relevance.

Other variables—such as sex (mutual information = 0.2251), pleural attachment (0.2203), and the detection of a nodule located within the same lung lobe (0.2137)—demonstrated a moderate impact on the model’s predictive performance. In contrast, the speculation feature (Yes/No) exhibited the lowest mutual information score (0.1712), highlighting its limited value in distinguishing EGFR mutation status.

These findings are further reinforced by the Spearman correlation analysis shown in Table 4, where all variables reached statistical significance (*p* < 0.05), and most displayed moderate to strong correlation intervals. Altogether, these results highlight the complementary value of clinical and CT-derived imaging features and justify their inclusion in the model’s feature selection process for robust EGFR mutation prediction.

#### 3.2.2. Radiomic Feature Selection and Correlation Analysis

Radiomic feature selection was conducted using FeatureWiz [49], a robust and versatile tool designed to identify the most informative predictors for building high-performance classification models. FeatureWiz integrates multiple selection strategies, including mutual information, recursive elimination, and model-based importance ranking, to evaluate each feature’s contribution to predictive accuracy. This process enables the elimination of irrelevant or redundant variables that may otherwise introduce noise or reduce model generalizability. All analyses were implemented using Python 3.7.6 and relevant open-source libraries.

From the initial pool of 1034 extracted radiomic features, a curated subset of 124 variables was selected based on their statistical significance and contribution to model performance. Within this reduced set, 20 radiomic features demonstrated a meaningful association with EGFR mutation status, as reflected by Spearman correlation coefficients ranging from approximately 0.11 to 0.26 (see Table 5).

Among these, the feature Exponential_Glrlm_Shortrunemphasis exhibited the highest observed correlation (r = 0.259), suggesting high discriminative potential and reinforcing its role as a promising radiomic biomarker for the non-invasive prediction of EGFR mutations in patients with NSCLC.

### 3.3. Performance Assessment Under Stratified Five-Fold Cross-Validation

#### 3.3.1. ROC Curve Analysis and Comparative Classification Performance

Receiver operating characteristic (ROC) analyses were performed for each predictive model. To avoid redundancy and ensure theoretical coherence, only the micro-averaged ROC curves with their 95% confidence intervals (CIs) are reported in Figure 4.

This approach provides a robust global summary of model discrimination ability while avoiding inconsistencies between per-class AUCs.

Among the tested models, Model 5, trained exclusively on the most relevant clinical, CT, and radiomic features, demonstrated the best overall performance. It achieved a micro-average AUC of 0.91 (95% CI: 0.81–1.00), confirming its robustness across resampling folds.

In parallel, additional classification metrics, namely precision, recall, F1-score, and overall accuracy, were calculated to further assess model performance. To ensure robust and unbiased evaluation, model performance was assessed using five-fold stratified cross-validation. This strategy preserved the class distribution between EGFR wild-type (*n* = 98) and EGFR mutant (*n* = 40) across all folds. The classification metrics were computed for each fold, and the final results were reported as mean ± standard deviation (SD) across the five folds. This comparative modeling approach allowed us to assess the relative contribution of each feature type (clinical, morphological (CT), and radiomic), as well as the impact of dimensionality reduction, through selection of the most relevant predictors. These values are reported in Table 6.

Cross-validated evaluation metrics confirmed robustness, with a recall of 0.92 ± 0.03, an accuracy of 0.88 ± 0.03, a precision of 0.93 ± 0.04, and an F1-score of 0.91 ± 0.02, indicating a well-balanced and reliable classification capability across both EGFR-Mutant and wild-type classes. When analyzed by class, the EGFR-WT subgroup achieved higher predictive values (precision: 0.93 ± 0.04, recall: 0.92 ± 0.03, F1-score: 0.91 ± 0.02), whereas the EGFR-Mutant subgroup obtained lower but clinically meaningful results (precision: 0.76 ± 0.05, recall: 0.71 ± 0.05, F1-score: 0.68 ± 0.04). The use of stratified five-fold cross-validation reduces sensitivity to small subgroup partitions, yielding more reliable and generalizable estimates for both EGFR-WT and EGFR-Mutant patients. While the model achieved higher performance in the EGFR-WT subgroup, the EGFR-Mutant subgroup results remain clinically relevant, supporting its use as a screening tool when invasive testing is limited. These results, detailed in Table 6, highlight the benefit of integrating optimized multimodal features in the prediction of EGFR mutation status in NSCLC patients.

All models were optimized through hyperparameter tuning, with final hyperparameters summarized in Appendix A. The corresponding confusion matrices for each model are provided in Appendix A.

#### 3.3.2. Decision Curve Analysis

Decision curve analysis (DCA) was performed under the stratified five-fold cross-validation framework to estimate the net benefit of each Random Forest model across different threshold probabilities. Figure 5 presents the fold-averaged DCA curves separately for training (a) and validation (b) folds. Model 5 consistently provided superior net benefit across clinically relevant thresholds. As shown in Figure 5, Model 5, which integrates a total of 11 keys features were selected, including seven radiomic variables, two CT morphological descriptors, and two clinical parameters, consistently outperformed the other models across a broad range of decision thresholds. The curve demonstrates that Model 5 consistently offers a superior net clinical benefit, especially across probability thresholds relevant to clinical decision making, reinforcing its applicability as a supportive tool for identifying NSCLC patients with a high likelihood of harboring EGFR mutations.

Model 5, built from the most informative combination of clinical, CT, and radiomic features, consistently demonstrates the highest net benefit across all thresholds, underscoring its superior potential for clinical decision making in EGFR mutation prediction.

#### 3.3.3. Statistical Comparison of AUCs Using the DeLong Test

To determine whether the observed differences in predictive performance were statistically meaningful, pairwise comparisons of the AUC values were performed using the DeLong test. As summarized in Table 7, the Combined—Selected CT, Clinical, and Radiomic Features model demonstrated significantly higher AUCs compared to the Clinical and CT—Full Feature Set, Radiomics—All Extracted Features, and Radiomics—Filtered Key Features models (*p* < 0.05), underscoring its superior discriminative capacity.

Although no statistically significant difference was found between the Combined model and the Clinical and CT—Selected Features Only model, the former consistently achieved the highest AUC values across all comparisons, further validating its effectiveness as the most robust model for EGFR mutation prediction.

As highlighted in Table 7, the Combined—Selected CT, Clinical, and Radiomic Features model achieved significantly higher AUCs compared with the Clinical and CT—Full Feature Set, Radiomics—All Extracted Features, and Radiomics—Filtered Key Features models (*p* < 0.05). While no significant difference was observed when compared with the Clinical and CT—Selected Features Only model (*p* = 0.201), the Combined model consistently achieved the highest AUCs, confirming it as the most robust configuration

### 3.4. Model Interpretability Using SHAP Analysis

To enhance the transparency and interpretability of the predictive model, the SHapley Additive exPlanations (SHAP) algorithm [48] was employed to quantify the influence of each feature on the model’s output in predicting EGFR mutation status. SHAP provides both global feature importance, highlighting the most influential variables across the entire dataset, and local explanations, offering patient-specific insights into individual predictions. Moreover, SHAP captures feature interaction effects, allowing a deeper understanding of how combinations of variables contribute to the model’s decision-making process. This approach supports the interpretability of complex machine learning models and facilitates their integration into clinical workflows.

#### 3.4.1. Global Feature Importance Analysis

To quantify each variable’s influence on model output, the mean SHAP scores from Model 5 were analyzed. Figure 6a highlights four dominant predictors: Age and Tobacco Use from the clinical domain, and Enhancement Pattern and Same-Lobe Nodule Presence from CT-based morphological features.

In addition, Figure 6b displays the global relevance of the selected radiomic features. Among these, seven demonstrated high predictive contribution. Particularly important were features extracted from the Gray-Level Dependence Matrix (GLDM), such as High Gray-Level Emphasis, and from the Gray-Level Size Zone Matrix (GLSZM), including Zone Entropy, both of which played a significant role in the model’s output.

These results confirm the strength of the selected feature set, which includes two clinical, two CT-based, and seven radiomic descriptors. This combination was shown to be crucial in achieving the high predictive performance observed in Model 5.

#### 3.4.2. Patient-Level Feature Impact Analysis

The SHAP algorithm was applied to evaluate the contribution of individual features to EGFR mutation prediction at the level of each patient. As shown in Figure 7a, clinical and CT features exhibited distinct patterns of influence on model decisions, with some variables contributing positively and others negatively to the predicted probability.

Among clinical features, tobacco use demonstrated a strong positive SHAP value, indicating that patients with higher tobacco exposure were more likely to be predicted as EGFR-mutation-positive. In contrast, age showed a marked negative influence, suggesting that younger patients were more frequently associated with EGFR mutations.

Among the CT-derived variables, enhancement pattern demonstrated a negative contribution in the SHAP analysis, suggesting that greater enhancement intensity was linked to a lower predicted probability of EGFR mutation. Similarly, a higher number of nodules located within the same lobe was associated with a decreased mutation likelihood. In contrast, sex and pleural retraction showed negligible influence on model predictions, as reflected by SHAP values near zero, aligning with their limited global relevance.

Figure 7b displays the individual impact of radiomic features on model predictions. Among these, High Gray-Level Emphasis (derived from the GLDM matrix) exhibited a strong positive contribution to EGFR mutation prediction, whereas Zone Entropy (extracted from the GLSZM matrix) showed a negative association, suggesting that greater textural irregularity was linked to a decreased likelihood of mutation. These findings underscore the complementarity between clinical, CT-derived, and radiomic information, and highlight how SHAP-based interpretability enables the identification of both intuitive and nuanced feature relationships influencing patient-specific outcomes.

#### 3.4.3. SHAP-Based Interaction Analysis of Predictive Features

SHAP interaction values were computed to explore how combinations of features jointly influence the model’s prediction of EGFR mutation status. This analysis provides deeper interpretability by identifying synergistic or conditional relationships between variables.

As shown in Figure 8a, the contribution of tobacco use to EGFR mutation prediction varies according to age. The predictive effect of smoking is most pronounced in younger patients, while it becomes less influential in older individuals, suggesting an age-dependent interaction.

In contrast, Figure 8b indicates that age-related predictive patterns remain consistent regardless of sex, reinforcing the previously observed minimal contribution of this variable in both global and local SHAP interpretations.

Notable interactions between clinical and imaging variables were observed. As illustrated in Figure 8c, the predictive contribution of a nodule located in the same lobe was more pronounced in patients with a history of smoking. This finding highlights a potentially meaningful association between lifestyle-related factors and radiological indicators in the context of EGFR mutation prediction.

Radiomic features revealed additional interaction effects. In Figure 8d, the feature Wavelet-HLL_GLCM_Imc1 exerts a stronger predictive influence when Wavelet-HLL_GLDM_LowGrayLevelEmphasis values are low. Similarly, Figure 8e shows that Wavelet-HLL_GLDM_ShortRunLowGrayLevelEmphasis and Wavelet-HLL_GLCM_Imc1 jointly modulate model output. Finally, Figure 8f illustrates that the combined effect of Wavelet-HLL_GLCM_Imc1 and Wavelet-HLL_FirstOrder_Mean enhances prediction, demonstrating interaction between textural and intensity-based radiomic descriptors.

Overall, these findings highlight the multidimensional complexity of EGFR mutation prediction and underscore the usefulness of SHAP interaction analysis in revealing nuanced interdependencies among clinical, morphological, and radiomic features.

## 4. Discussion

### 4.1. Key Outcomes and Relevance for Clinical Practice

This study presents a comprehensive machine learning approach that integrates clinical data, CT-derived morphological indicators, and radiomic signatures to enable the non-invasive prediction of EGFR mutation status in individuals diagnosed with non-small cell lung cancer (NSCLC). Leveraging a Random Forest (RF) classification approach, five predictive models were evaluated based on distinct combinations of features. Feature selection played a central role in optimizing performance, using a three-step pipeline that included mutual information analysis, Spearman correlation filtering, and reduction of redundancy. For radiomics, additional selection was performed using the FeatureWiz algorithm to retain only the most discriminative variables among the 1034 initially extracted features.

Among the models evaluated, Model 5, which was trained solely on the most predictive feature subset, including two clinical variables (Age, Tobacco use), two CT morphological characteristics (Enhancement pattern, Presence of nodules in the same lobe), and seven selected radiomic features, achieved the strongest predictive performance, reflected by an AUC of 0.91 (95% CI: 0.81–1.00). It also delivered high values in other classification metrics, including precision, recall, and F1-score. To improve model transparency, SHapley Additive exPlanation (SHAP) was applied, offering insight into both global and patient-specific feature contributions, and revealing interactions among key variables.

This approach highlights the potential of non-invasive imaging biomarkers to serve as surrogates for molecular testing, particularly in clinical scenarios where tissue sampling is limited or risky [27,28,44,45]. Early and accurate identification of patients with EGFR mutations may support the timely initiation of tyrosine kinase inhibitors (TKIs), which have demonstrated improved outcomes in mutation-positive NSCLC patients [50,51,52].

### 4.2. Current Methodological Constraints and Future Research Directions

Although the outcomes are encouraging, it is important to recognize certain methodological constraints and unresolved issues that may affect the model’s generalizability and clinical applicability. First, the study design is retrospective and single-center, which limits the external validity of the model. To establish generalizability and clinical utility, prospective validation in multi-institutional cohorts is required. Additionally, although SMOTE-based oversampling and data augmentation techniques were used to mitigate class imbalance and limited sample size, these strategies may not fully capture the variability found in real-world populations.

A further limitation is the modest size of the EGFR-Mutant subgroup (*n* = 40). This reflects the real-world difficulty of prospectively collecting large balanced datasets in oncology, particularly in North Africa. While this small subgroup size makes single-split evaluations unstable, the use of stratified cross-validation allowed us to obtain more robust and reproducible estimates. Future multi-center studies with larger populations will be essential to confirm reproducibility.

An additional limitation concerns the SHAP analysis of tobacco use, which paradoxically emerged as a positive contributor to predicting EGFR mutation. This finding contradicts established clinical evidence, as EGFR mutations are more prevalent among non-smokers. After verifying the dataset, no variable misencoding was identified, suggesting that this apparent contradiction reflects cohort-specific bias, given the relatively small number of mutant cases (*n* = 40). This emphasizes that SHAP contributions in small datasets may capture sample-specific distributions rather than general biological associations. Future validation on larger and more diverse cohorts will be required to determine whether this effect persists or diminishes.

The integration of radiomics into clinical workflows remains a challenge due to the lack of standardization in radiomic feature extraction and preprocessing protocols, which affects reproducibility across platforms and scanners [52]. Furthermore, clinical integration would require seamless interoperability with existing hospital information systems and radiology workflows, which may involve significant technological and organizational adaptations.

Future work should aim to meet the following objectives:

Validate the model in diverse populations with different genetic and demographic backgrounds; assess the longitudinal predictive performance across time points and treatment phases; explore the integration of deep-learning-based radiomics or hybrid approaches combining handcrafted and deep features.

Furthermore, while SHAP was used in this study primarily for post hoc model interpretability, its integration into the feature selection pipeline could provide an additional validation layer for identifying the most predictive variables. Comparing SHAP-based feature selection with our current statistical and algorithmic methods may enhance robustness and offer deeper insights into the biological relevance of selected radiomic descriptors. We recognize this as an important avenue for future work.

In addition, future research should compare the performance of Random Forest classifiers with other machine learning algorithms, such as support vector machines, gradient boosting, and deep learning architectures, particularly when larger multicenter datasets become available. This would allow a more comprehensive benchmarking of classification strategies and enhance the generalizability of the proposed framework.

Another methodological consideration concerns the choice of learning framework. Although deep learning approaches, such as convolutional neural networks (CNNs), have demonstrated excellent performance in image-based predictive tasks, their application typically requires large-scale datasets to avoid overfitting. Given the relatively modest cohort size in the present study (*n* = 138), we opted for Random Forest classifiers, which are more suitable for small and heterogeneous datasets while offering interpretable outputs for clinical translation. Nevertheless, future work with larger, multi-institutional cohorts will investigate CNN-based architectures and hybrid models, which combine handcrafted radiomic features with deep feature representations, in order to benchmark their performance against our current machine learning pipeline.

### 4.3. Contribution, Relevance, and Context

To our knowledge, this is the first study conducted in Morocco that investigates the non-invasive prediction of EGFR mutation status in NSCLC patients through the application of machine learning techniques combined with multimodal-imaging-derived biomarkers. Our findings are particularly relevant in a context where biopsy access is often limited, and where genetic profiling may not be routinely available.

While our cohort’s EGFR mutation prevalence (28.98%) is lower than rates reported in East Asian populations, it remains aligned with those observed in Middle Eastern, North African, and European cohorts [53]. This reinforces the relevance of developing region-specific predictive models that account for population-specific clinical and genetic profiles.

Importantly, the study provides a foundational framework for future research and clinical implementation of AI-assisted decision support tools in Moroccan oncology settings. By combining CT-derived radiomics with accessible clinical data, this model offers a promising direction toward personalized, non-invasive diagnostics in lung cancer care, particularly for patients for whom tissue sampling is contraindicated or unfeasible.

## 5. Conclusions

This work presents an automated and robust approach for the non-invasive prediction of EGFR mutation status in patients with non-small cell lung cancer (NSCLC). By combining clinical data, CT-based morphological characteristics, and quantitative radiomic features, we constructed a machine learning model capable of reliably identifying patients with a high probability of carrying EGFR mutations.

Such a predictive tool may serve as a valuable adjunct in clinical workflows, especially in scenarios where histopathological confirmation is not feasible or molecular testing is delayed. By facilitating the early identification of patients eligible for tyrosine kinase inhibitor (TKI) therapy, this approach holds promise for improving the timeliness and personalization of NSCLC treatment.

Among the five models assessed, the best performance was achieved by the one trained on a targeted selection of features, comprising two clinical variables, two CT morphological attributes, and seven radiomic descriptors. This optimized feature combination led to notable improvements across all performance indicators, including enhanced precision, recall, and overall classification accuracy.

This method demonstrates significant promise as a clinical decision support tool, especially in scenarios where tissue sampling is constrained or molecular diagnostics are inaccessible. Facilitating the rapid identification of candidates for TKI therapy may enhance the timeliness and individualization of care for NSCLC patients.

Recommendations and Future Research: To strengthen clinical translation, future studies should focus on multi-center validation with larger and more diverse patient cohorts, particularly including non-adenocarcinoma NSCLC subtypes. Prospective studies are also warranted to evaluate real-world performance and integration into routine oncology workflows. Additionally, the combination of radiomics with liquid biopsy markers and advanced deep learning approaches could further improve predictive accuracy and generalizability. Complementary comparisons with alternative machine learning algorithms such as XGBoost and artificial neural networks (ANNs) are also recommended to benchmark performance and to enhance model robustness.

## Figures and Tables

**Figure 1 arm-93-00039-f001:**
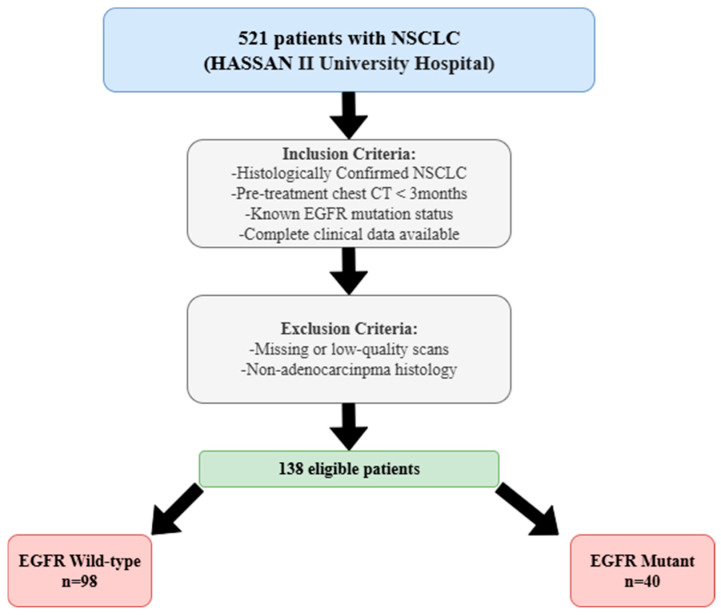
Diagram outlining the step-by-step selection of NSCLC patients according to established inclusion and exclusion criteria, resulting in the final cohort used for model training and validation.

**Figure 2 arm-93-00039-f002:**
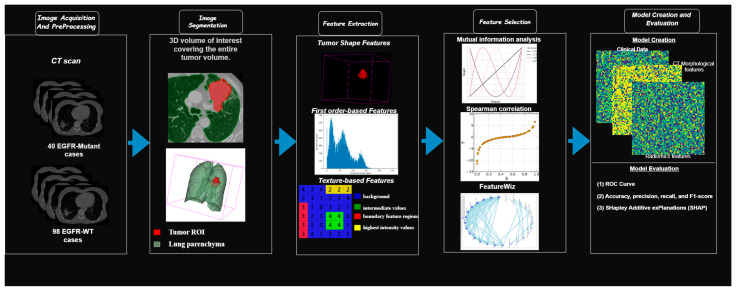
Overview of the study workflow illustrating the sequential stages from CT image acquisition and tumor segmentation to radiomic feature extraction, feature selection, model training, and performance evaluation.

**Figure 3 arm-93-00039-f003:**
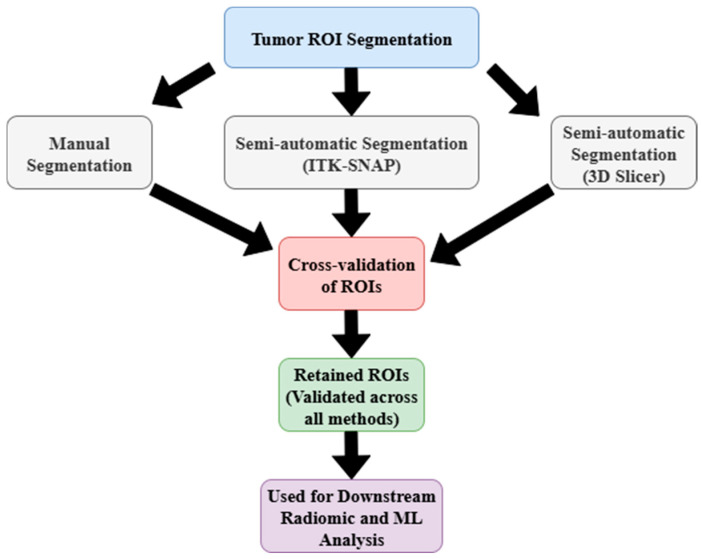
Workflow for tumor ROI segmentation and validation using manual and semi-automatic methods.

**Figure 4 arm-93-00039-f004:**
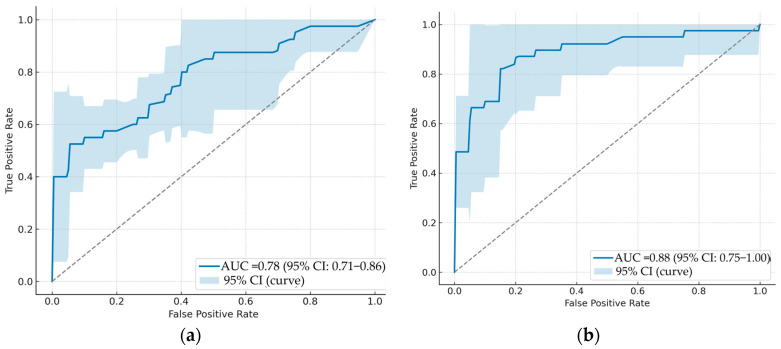
Micro-averaged receiver operating characteristic (ROC) curves with 95% confidence intervals for the five Random Forest models: (**a**) Model 1, (**b**) Model 2, (**c**) Model 3, (**d**) Model 4, (**e**) Model 5.

**Figure 5 arm-93-00039-f005:**
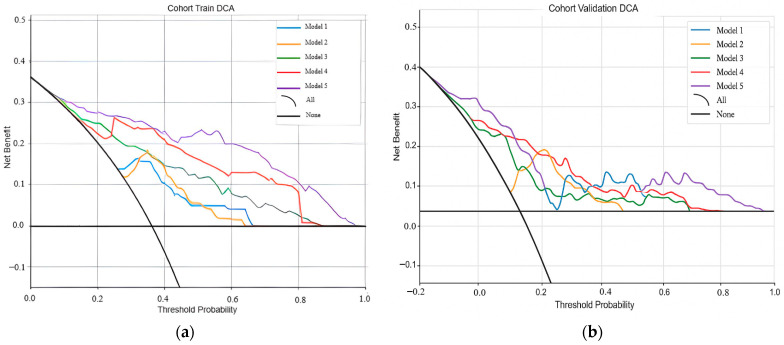
Decision curve analysis (DCA) of the five Random Forest models under stratified five-fold cross-validation: (**a**) fold-averaged training curves, (**b**) fold-averaged validation curves.

**Figure 6 arm-93-00039-f006:**
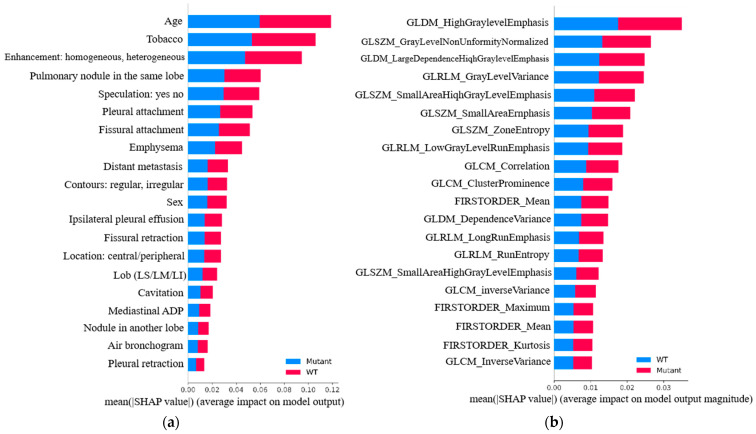
SHAP-based global feature importance for Model 5: (**a**) clinical and CT variables, (**b**) selected radiomic features.

**Figure 7 arm-93-00039-f007:**
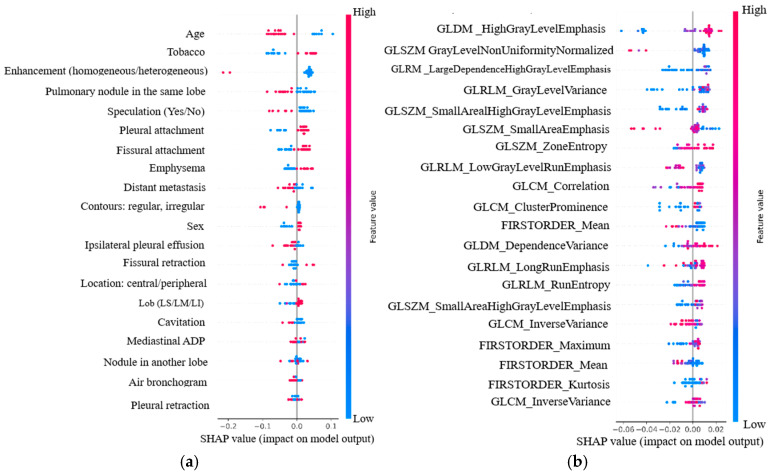
SHAP-derived feature contributions for EGFR mutation prediction in Model 5: (**a**) clinical and CT variables, (**b**) radiomic features.

**Figure 8 arm-93-00039-f008:**
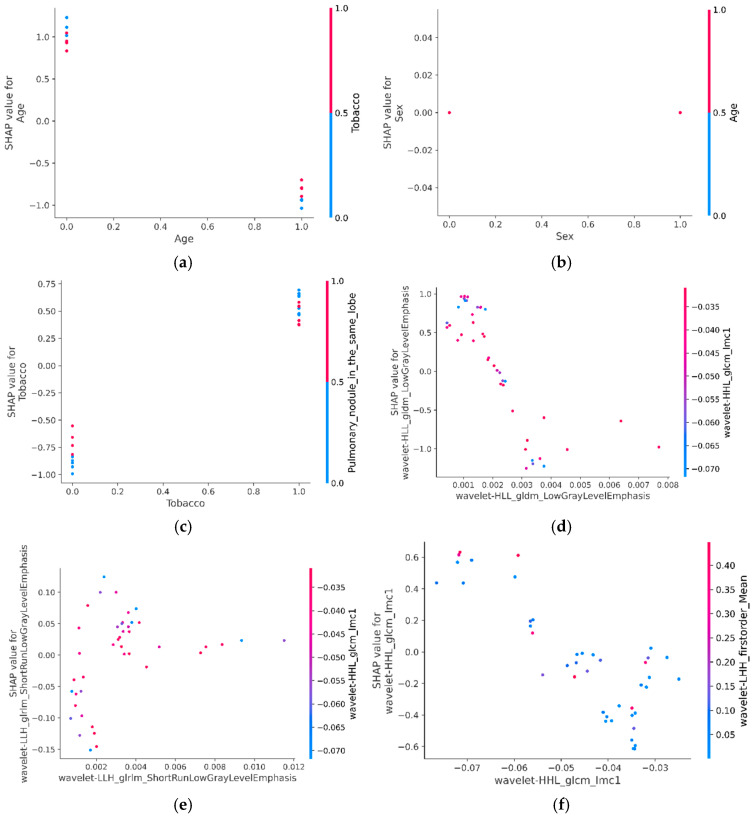
SHAP-based interaction plots for Model 5: (**a**) age and tobacco use, (**b**) sex and age, (**c**) tobacco use and CT nodule presence, (**d**–**f**) radiomic feature interactions.

**Table 1 arm-93-00039-t001:** Dataset characteristics used in this study.

Dataset Name	Source	Histological Subtype	Number of Patients	EGFR Status Distribution	Acquisition Period
NSCLC Cohort—Hassan II University Hospital	Hassan II University Hospital, Fez, Morocco	Adenocarcinoma	138	EGFR wild-type (*n* = 98), EGFR mutant (*n* = 40)	2022–2024

**Table 2 arm-93-00039-t002:** Composition of datasets used for training the Random Forest models.

Model Description	Feature Set Description
Clinical and CT—Full Feature Set	Includes all available clinical and CT morphological features without feature selection.
Clinical and CT—Selected Features Only	Includes only the highly relevant features extracted from clinical data and CT morphological features identified through feature selection.
Radiomics—All Extracted Features	Comprises the full set of extracted radiomic features without dimensionality reduction.
Radiomics—Filtered Key Features	Incorporates only the most informative radiomic features selected through statistical and algorithmic filtering.
Combined—Selected CT, Clinical, and Radiomic Features	Combines the most relevant CT, clinical, morphological, and radiomic features into an integrated predictive dataset.

**Table 3 arm-93-00039-t003:** Mutual information scores for selected clinical and CT features.

Feature	Mutual Information Score
Enhancement (homogeneous/heterogeneous)	0.4517
Tobacco	0.3289
Age	0.2584
Pulmonary nodule in the same lobe	0.2137
Pleural attachment	0.2203
Sex	0.2251
Speculation (Yes/No)	0.1712

**Table 4 arm-93-00039-t004:** *p*-values and 95% confidence intervals (Spearman correlation).

Feature	*p*-Value	95% CI (Spearman)
Enhancement (homogeneous/heterogeneous)	0.000	[0.442, 0.999]
Tobacco	0.000	[0.344, 0.982]
Age	0.010	[0.219, 0.730]
Pulmonary nodule in the same lobe	0.021	[0.197, 0.648]
Pleural attachment	0.009	[0.222, 0.742]
Sex	0.0006	[0.232, 0.778]
Speculation (Yes/No)	0.043	[0.214, 0.608]
Enhancement (homogeneous/heterogeneous)	0.000	[0.442, 0.999]

**Table 5 arm-93-00039-t005:** Radiomic features associated with EGFR mutation (Spearman correlation).

Feature	Spearman Correlation Coefficient (r)
Exponential_Glrlm_Shortrunemphasis	0.259262
Wavelet-HHH_Glszm_Smallareaemphasis	0.256393
Wavelet-HLH_Firstorder_Mean	0.257096
Wavelet-LHH_Firstorder_Mean	0.244431
Wavelet-HHL_Gldm_Smalldependencelowgraylevelemphasis	0.222836
Wavelet-HHL_Firstorder_Mean	0.217945
Wavelet-LLL_Glcm_Imc1	0.215338
Wavelet-HHL_Glcm_Imc1	0.211136
Square_Ngtdm_Strength	0.209912
Log-Sigma-2-0-Mm-3D_Glrlm_Shortrunlowgraylevelemphasis	0.204865
Wavelet-LHH_Glszm_Smallareaemphasis	0.198212
Log-Sigma-2-0-Mm-3D_Glszm_Zonevariance	0.162736
Log-Sigma-3-0-Mm-3D_Glszm_Graylevelnonuniformitynormalized	0.155103
Wavelet-LHH_Glcm_Imc1	0.170933
Wavelet-HLL_Gldm_Lowgraylevelemphasis	0.168319
Original_Shape_Elongation	0.136899
Exponential_Firstorder_90Percentile	0.125991
Wavelet-LLH_Glrlm_Shortrunlowgraylevelemphasis	0.120218
Wavelet-LLH_Glcm_Imc1	0.118433
Wavelet-HHH_Glszm_Lowgraylevelzoneemphasis	0.113427

**Table 6 arm-93-00039-t006:** Performance of Random Forest models with mean ± SD from five-fold cross-validation.

Model	Class	Precision (Mean ±SD)	F1-Score (Mean ±SD)	Recall (Mean ±SD)	Accuracy (Mean ±SD)
Model 1: Random Forest model trained using all available clinical and CT morphological features without feature selection.	EGFR-WT	0.84 ± 0.02	0.83 ± 0.05	0.83 ± 0.04	0.77 ± 0.04
	EGFR-Mutant	0.58 ± 0.06	0.59 ± 0.07	0.58 ± 0.06	
	Macro-average	0.71 ± 0.05	0.72 ± 0.06	0.71 ± 0.05	
	Weighted average	0.78 ± 0.04	0.79 ± 0.03	0.78 ± 0.04	
Model 2: Random Forest model trained using only the most relevant clinical and CT morphological features identified through feature selection.	EGFR-WT	0.87 ± 0.03	0.85 ± 0.04	0.86 ± 0.03	0.82 ± 0.03
	EGFR-Mutant	0.73 ± 0.05	0.72 ± 0.06	0.72 ± 0.05	
	Macro-average	0.80 ± 0.05	0.79 ± 0.04	0.79 ± 0.04	
	Weighted average	0.84 ± 0.04	0.83 ± 0.03	0.83 ± 0.03	
Model 3: Random Forest model trained on the full set of extracted radiomic features without dimensionality reduction.	EGFR-WT	0.80 ± 0.05	0.79 ± 0.05	0.79 ± 0.05	0.72 ± 0.04
	EGFR-Mutant	0.46 ± 0.07	0.49 ± 0.07	0.47 ± 0.06	
	Macro-average	0.63 ± 0.05	0.64 ± 0.06	0.63 ± 0.06	
	Weighted average	0.76 ± 0.06	0.75 ± 0.05	0.75 ± 0.05	
Model 4: Random Forest model trained on a subset of radiomic features selected for high relevance.	EGFR-WT	0.86 ± 0.04	0.84 ± 0.02	0.85 ± 0.04	0.80 ± 0.05
	EGFR-Mutant	0.65 ± 0.06	0.67 ± 0.06	0.66 ± 0.06	
	Macro-average	0.77 ± 0.02	0.76 ± 0.05	0.76 ± 0.05	
	Weighted average	0.84 ± 0.05	0.83 ± 0.04	0.83 ± 0.04	
Model 5: Random Forest model trained on the most informative combination of clinical, CT, and radiomic features.	EGFR-WT	0.93 ± 0.04	0.91 ± 0.02	0.92 ± 0.03	0.88 ± 0.03
	EGFR-Mutant	0.76 ± 0.05	0.68 ± 0.04	0.71 ± 0.05	
	Macro-average	0.84 ± 0.04	0.80 ± 0.04	0.82 ± 0.04	
	Weighted average	0.90 ± 0.03	0.89 ± 0.03	0.89 ± 0.02	

**Table 7 arm-93-00039-t007:** Pairwise comparison of AUCs between Random Forest models using DeLong test *p*-values.

	Clinical and CT—Full Feature Set	Clinical and CT—Selected Features Only	Radiomics—All Extracted Features	Radiomics—Filtered Key Features	Combined—Selected CT, Clinical, and Radiomic Features
Clinical and CT—Full Feature Set	1.000	0.027	0.688	0.087	0.004
Clinical and CT—Selected Features Only	0.040	1.000	0.064	0.461	0.201
Radiomics—All Extracted Features	0.681	0.052	1.000	0.201	0.004
Radiomics—Filtered Key Features	0.085	0.474	0.192	1.000	0.050
Combined—Selected CT, Clinical, and Radiomic Features	0.000	0.206	0.006	0.046	1.000

## Data Availability

The data supporting the findings of this study are available from the corresponding authors upon reasonable request.

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
