# Peer review of "NSCLC EGFR Mutation Prediction via Random Forest Model: A Clinical–CT–Radiomics Integration Approach"

_arm, 2025, doi:10.3390/arm93050039_

Round 1
Reviewer 1 Report
Comments and Suggestions for Authors
Reviewer Recommendation: Major Revision
Overall Evaluation
This study integrates clinical data, CT morphological features, and radiomic features with a random forest classifier and SHAP-based interpretability analysis to predict EGFR mutation status in NSCLC patients. The research topic holds clinical significance and aligns with trends in precision oncology and explainable AI. However, several shortcomings require revision before publication.
- 1.Lack of Model Hyperparameter Tuning
Random forest performance critically depends on hyperparameters (e.g., n_estimators, max_depth, min_samples_leaf). The authors compared feature combinations (Table 1) but did not justify parameter selection or report final values (e.g., Model 5’s tree count/depth). Default parameters risk underfitting (insufficient depth) or overfitting (excessive depth), undermining Table 5’s reliability (e.g., Accuracy=0.87).
Recommendations:
(1) Add hyperparameter optimization (e.g., 5-fold cross-validated grid search) specifying the target metric (e.g., AUC/F1-score).
(2) List all models’ final hyperparameters in Materials and Methods or supplements.
- Inadequate Model Validation
A single train-test split (96/42 samples) was used without cross-validation (CV). For small datasets (mutant subgroup: n=40), this risks performance overestimation due to randomness.
Recommendations:
(1) Implement K-fold CV (5–10 folds), reporting mean performance ± SD.
(2) Use stratified CV to preserve class ratios if sample size constraints persist.
- Contradiction Between Abstract and Main Text
The abstract/results claim the "best model" achieves *"Precision 0.90, Recall 0.94, F1-score 0.91, Accuracy 0.87"*—but Table 5 shows these are EGFR-WT results. The EGFR-Mutant class results (*Precision 0.71, Recall 0.5, F1-score 0.59*) were misrepresented, misleading readers about minority-class performance.
Recommendations: Clarify results, explicitly stating EGFR-Mutant performance.
- Anomalous F1-Score in Table 5 (Model 2)
For EGFR-Mutant: *Precision=0.85, Recall=0.75* → Theoretical F1-score = 2×(0.85×0.75)/(0.85+0.75) ≈ 0.80, but Table 5 reports 0.33 (severe calculation/formatting error).
- Anomalous Macro-Average Precision in Table 5 (Model 2)
Model 2’s macro-average precision=0.90 exceeds the mean of EGFR-WT precision (0.80) and EGFR-Mutant precision (0.85), indicating calculation error.
Recommendation: Recalculate all metrics, correct Table 5, and provide confusion matrices or TP/FP/FN/TN values for verification.
- Inconsistencies in ROC Curves and AUC (Figure 3)
(1) In binary classification, ROC curves for Class 0 and Class 1 should exhibit mirror symmetry about the diagonal with identical AUC when using identical predicted probabilities. Figure 3 redundantly reports per-class, macro, and micro AUCs (e.g., Model 5: macro=0.90, micro=0.89, AUC=0.88), violating theory.
(2) In Figure 3(e), Class 0 and Class 1 curves have different shapes but identical AUCs—theoretically implausible unless using distinct scoring/plotting methods.
Recommendations:
- or each model, show one ROC curve(micro/macro) with 95% CI.
- If per-class curves are retained, move to supplements and explain theoretical equivalence/observed discrepancies.
- Interpretation Paradox in SHAP Results
SHAP indicates "smoking" strongly positively contributes to predicting EGFR mutation, contradicting clinical consensus (mutations are more prevalent in non-smokers). This may arise from label misencoding, confounding, or sample bias.
Recommendation: Verify variable encoding, conduct stratified analysis, and explain this anomaly in the Discussion.
- Additional Issues
(1) Improve Figure 2 clarity (text is partially obscured).
(2) Correct duplicate numbering in Figure 3.
(3) Supplement CT scan parameters (kVp, mAs, slice thickness, reconstruction kernel, contrast phase).
(4) Avoid terms like "strong correlation" for low coefficients (max |r|≈0.26).
(5) Remove or expand the "2D deep learning" method mention (unaddressed in results).
(6) Fix Table 5 formatting (duplicate "F1-Score" column).
(7) Add confusion matrices for both classes in Table 5.
(8) Correct Figure 3 numbering.
Revision Requirement
The revised manuscript must resolve all above issues and provide:
- Point-by-point revision descriptions.
- Supporting materials (e.g., cross-validation results, corrected tables/figures, confusion matrices, hyperparameter tuning details).
Acceptance is contingent on rigorous verification of these amendments.
Reviewer 2 Report
Comments and Suggestions for Authors
This study developed an interpretable machine learning model to non-invasively predict EGFR mutation status in non-small cell lung cancer (NSCLC) patients using clinical data, CT morphological features, and radiomic characteristics. However, the manuscript requires improvements in writing, particularly in the introduction and methodology sections.
Comments:
- The title and abstract contain several acronyms that may not be familiar to general readers. Please define them clearly.
- In the introduction, there is no evident research gap compared with existing works [36–38]. The authors should provide a more detailed discussion of the limitations of previous studies and explain what their work contributes to addressing those gaps.
- Section 2.1 should be restructured. Instead of simply listing the steps, please elaborate in paragraph form to describe the step-by-step framework (e.g., from image acquisition to model evaluation) as illustrated in Figure 1.
- The dataset description, including details on its samples and features, is unclear. The data are imbalanced, with 40 mutant cases and 98 WT cases. How was the dataset split? Did the authors use cross-validation to minimize bias?
- The authors should analyze the segmentation performance and provide visualizations of the results in the experimental section.
- All figures in the manuscript appear blurry. Please improve their resolution for better readability.
- The limitations of this study should be discussed at the end of the conclusion, along with potential directions for future work.
Reviewer 3 Report
Comments and Suggestions for Authors
Thank you for providing opportunity to review the manuscript
Please find my below comments for improvement
Introduction and methodology should includes about the types of NSCLC which were included in the study? whether its adenocarcinoma or squamous cell carcinoma
Confusion matrix for ML models are missing, needs to be provided
Why only random forest classifier was used? what about other ML classifiers
Region of tumor segmentation needs to mentioned in methodology
Reviewer 4 Report
Comments and Suggestions for Authors
This research is useful. However, there are suggestions for improvement from the author:
1) In the research methodology section, the author should provide details about the dataset used in a table format, such as data name, source, quantity, and date of data acquisition.
2) Figure 2 is low resolution. The author should improve the image resolution to make it clearer.
3) Section 2.1 provides an overview of the research methodology. Section 2.1 is not required; instead, Section 2.2 should be changed to 2.1. Furthermore, the subsections should align with the overall research steps.
4) Figure 3 is low resolution. The author should improve the image resolution to make it clearer.
5) The author should summarize the training parameter values in a table format for readers to use.
6) Section 3.1 should present the methodology instead of the results.
7) This research lacks a visual representation of the prediction results. The author should add more to enhance the credibility of the research.
8) The conclusion should include recommendations and future research.
Reviewer 5 Report
Comments and Suggestions for Authors
- what is EGFR? it's both used in title, highlights and abstract but no open form is given. Should all we have to know?
- ok, the definetion is given at line 55 but you must give it within the first usage
- line 58. is the writing of "exon" is true? not sure if I've heard it before?
- line 120-121. again lots of abb. but an open form? you know the rule, you have to give its open form in the first usage not at the last page
- you mentioned something called "3d slicer", but what is it and why it is important to use it?
- section 3.1. why using different train-test ratio for the egfr-mutated and egfr-wild cases? you have to use same ratio to prevent bias
- line 264-294. it's not so good to use all bullet /numbered list. it makes the paper structure really bad to read
- for spearman why p>0,85? explain better the selection of the value
- "(see Sorted and Adjusted Radiomic Feature Correlations table)" why not simply giving the table number?
- why only using random forest? it's really hard to understand without adding some other methods to compare? for example just add some basic ANN and especially XGBoost!
- table 6, you have to explain it better. which model is statistically better? p values should be added
- if you use SHAP, you also need to use it for feature selection to compare against all the features used in the model?
Reviewer 6 Report
Comments and Suggestions for Authors
In this study, a non-invasive, interpretable machine learning model is presented for predicting EGFR mutation status in NSCLC using clinical data, CT morphological characteristics, and radiomic features. The strength of the study lies in the successful results obtained by considering different features in the proposed model and adding interpretability to the model. However, for the study to contribute more to those conducting research in this field, the following points should be clarified.
1) The study considered a machine learning-based approach. However, rule-based approaches such as fuzzy logic and deep learning approaches such as convolutional neural networks are also available in the literature. In this context, why was a deep learning approach, which can provide better performance when compared to the machine learning approach, not preferred in the study?
2) What is the exact reason for choosing the random forests approach in the study? Why was no information provided about this approach, even briefly?
3) Why were more modern approaches such as CutMix, MixUp, and CutOut not used instead of classical approaches in the 2D data augmentation process?
4) Why was no information provided about the computational complexity of the proposed model?
Round 2
Reviewer 1 Report
Comments and Suggestions for Authors
Second Review Comments
Title of the Manuscript: [Prediction of EGFR Mutations in NSCLC via Random Forest Model: A Clinical-CT-Radiomics Integration Approach]
Journal: [Advances in Respiratory Medicine]
Review Recommendation: Reject
- Overall Evaluation
This paper describes the prediction of EGFR mutations in non-small cell lung cancer based on the random forest model. The revised version has made some adjustments to the suggestions proposed in the original version, but still has significant issues. It is recommended to reject the manuscript.
- Existing Issues
2.1 Inconsistency Between Supplementary Material and Main Text
This article included a total of 138 patients as the data set: 98 patients were diagnosed as wild-type EGFR patients, and 40 patients had activated EGFR mutations. Obviously, this is a typical binary classification prediction.
Table 6 claims to present the performance indicators of the test set (n=42), but the confusion matrix provided in Supplementary Table S2 is based on all 138 patients. This leads to confusion in data presentation and seriously undermines the credibility of the results. Moreover, the display of the confusion matrix in Supplementary Table S2 is not in the conventional format, causing confusion in this supplementary material.
2.2 Incomplete Presentation of Results
(1) The description of the result in the Highlights section is "AUC of 0.91", while according to the description in the main text, this result actually represents the prediction result for the EGFR-WT category;
(2) In the abstract, the description of the result is "The best model: Precision 0.90, Recall 0.94, F1-score 0.91, Accuracy 0.87". As can be seen from Table 6, this is also the prediction result for the EGFR-WT category.
The descriptions of the results in the above two places only showcase the better result for the EGFR-WT category, without presenting the prediction results for the EGFR-Mutant category. Selective reporting of high indicators is being done. The abstract should comprehensively reflect the performance of both categories.
2.3 Significant Changes in Key Data
In the initial draft, Table 5 (equivalent to Table 6 in this manuscript) of Model 5 for the EGFR-Mutant category showed a result of Precision 0.71, Recall 0.5, and F1-score 0.59. However, in this manuscript, Table 6 of Model 5 for the EGFR-Mutant category showed a result of Precision 0.76 ± 0.05, Recall 0.71 ± 0.05, and F1-score 0.68 ± 0.04. The recall value changed from 0.5 to [0.66, 0.76]. This significant change in key data results has led to a substantial performance improvement without altering the dataset, features, or model structure. Such a remarkable performance enhancement lacks a reasonable explanation and raises doubts about the reproducibility of the results or the stability of the data processing procedure.
In conclusion, it is recommended to reject.
Reviewer 2 Report
Comments and Suggestions for Authors
It has been significantly improved
Author Response
Comments and Suggestions for Authors:
It has been significantly improved
Response:
We sincerely thank the reviewer for the positive feedback and for acknowledging the significant improvements in our revised manuscript. We highly appreciate the reviewer’s constructive input, which helped us to enhance the quality and clarity of our work.
Reviewer 3 Report
Comments and Suggestions for Authors
All my comments were addressed
Author Response
Comments and Suggestions for Authors:All my comments were addressed
Response:
We would like to thank the reviewer for confirming that all the comments have been addressed. We are grateful for the reviewer’s careful assessment and constructive feedback, which have greatly improved the manuscript.
Reviewer 4 Report
Comments and Suggestions for Authors
-
Author Response
Comments and Suggestions for Authors :-
Response:
We thank the reviewer for the evaluation of our work. We appreciate the reviewer’s time and effort in reviewing our manuscript.
Reviewer 5 Report
Comments and Suggestions for Authors
- it seems there is a confusing situation at the title of table 5,6, 7 and fig 4-8. it does not seems a title in my opinion? just give a short title and give those explanations further in the text as a reference to related table/figure
- line 458: missing table number and title?
Round 3
Reviewer 1 Report
Comments and Suggestions for Authors
- Overall Evaluation
The revised supplementary materials provided the final optimized hyperparameters for each random forest model in Supplementary Table S1. However, there were significant adjustments to the key data results from version 1 to version 3, which is suspicious. We suggest rejection of the manuscript. While you can also provide us the information about how did you update this data?
- Critical Issues
2.1 Questionable Key Results
In the first version, Model 5 in Table 5 (equivalent to Table 6 in the revised manuscript) demonstrated the following performance metrics for the EGFR-Mutant class: Precision 0.71, Recall 0.5, and F1-score 0.59. However, in the second and third revisions, Model 5 in Table 6 reported significantly improved results: Precision 0.76 ± 0.05, Recall 0.71 ± 0.05, and F1-score 0.68 ± 0.04. Notably, the Recall value increased from 0.5 to a range of [0.66, 0.76], representing a 16% improvement, as illustrated in Figure 1. This substantial enhancement occurred without any modifications to the methodology or dataset, and the subsequent revisions failed to provide corresponding data to support these changes. Such unexplained alterations render the modifications highly questionable.
